# MONITORING LATENT WORLD STATES IN LANGUAGE MODELS WITH PROPOSITIONAL PROBES

**Jiahai Feng**,[*] **Stuart Russell & Jacob Steinhardt**
UC Berkeley

## ABSTRACT

Language models (LMs) are susceptible to bias, sycophancy, backdoors, and other tendencies that lead to unfaithful responses to the input context. Interpreting internal states of LMs could help monitor and correct unfaithful behavior. We hypothesize that LMs faithfully represent their input contexts in a latent world model, and we seek to extract these latent world states as logical propositions. For example, given the input context "Greg is a nurse. Laura is a physicist.", we aim to decode the propositions $\mathsf{WorksAs}(\mathsf{Greg}, \mathsf{nurse})$ and $\mathsf{WorksAs}(\mathsf{Laura}, \mathsf{physicist})$ from the model's internal activations. To do so we introduce *propositional probes*, which compositionally extract lexical concepts from token activations and bind them into propositions. Key to this is identifying a *binding subspace* in which bound tokens have high similarity ($\mathsf{Greg} \leftrightarrow \mathsf{nurse}$) but unbound ones do not ($\mathsf{Greg} \nleftrightarrow \mathsf{physicist}$). Despite only being trained on linguistically simple English templates, we find that propositional probes generalize to inputs written as short stories and translated to Spanish. Moreover, in three settings where LMs respond unfaithfully to the input context—prompt injections, backdoor attacks, and gender bias— the decoded propositions remain faithful. This suggests that LMs often encode a faithful world model but decode it unfaithfully, which motivates the search for better interpretability tools for monitoring LMs.

## 1 INTRODUCTION

Language models (LMs) may produce responses unfaithful to the input context in many situations: they can be misled by irrelevant examples (Anil et al., 2024; Halawi et al., 2023), learn unintended tendencies (Sharma et al., 2023) and biases (Blodgett et al., 2020; Liang et al., 2022), and are vulnerable to adversarial attacks on their prompts (Perez & Ribeiro, 2022) and training data (Wallace et al., 2020; Hubinger et al., 2024). This makes it important to be able to identify and correct when language models are unfaithful.

To address unfaithfulness, one promising approach is to interpret LMs' internal states (Christiano et al., 2021; Viégas & Wattenberg, 2023). A common hypothesis is that LMs internally represent the input context in a latent world model (Li et al., 2021; 2022), which could remain faithful even if the LM outputs falsehoods (Mallen & Belrose, 2023). This might happen, for instance, if unfaithful tendencies affect how the model behaves in accordance to its beliefs, but not the beliefs themselves.

In this work, we introduce *propositional probes*, which extract symbolic latent world states as logical propositions; we then show that these latent world states remain faithful in three adversarial settings. As an example of propositional probes, consider the input context "Alice lives in Laos. Bob lives in Peru." We would extract from internal activations the propositions $\{\mathsf{LivesIn}(\mathsf{Alice}, \mathsf{Laos}),$ $\mathsf{LivesIn}(\mathsf{Bob}, \mathsf{Peru})\}$ . In three adversarial settings, namely prompt injections, backdoors, and gender bias, we find that these extracted latent world states are more faithful than model outputs.

Our probes work by extracting lexical concepts from token activations, then binding them together to form propositions (Fig. 1) (Sec. 4). Since the space of propositions is exponentially large, our probes exploit compositionality by building propositions from their lexical constituents (e.g. composing $\mathsf{LivesIn}(\mathsf{Alice}, \mathsf{Laos})$ from $\mathsf{Alice}$ and $\mathsf{Laos}$). Key to enabling this is the "binding subspace" (Feng & Steinhardt, 2023), a subspace of activation space in which bound tokens (such as Alice and

---

[*]Correspondence to fjiahai@berkeley.edu

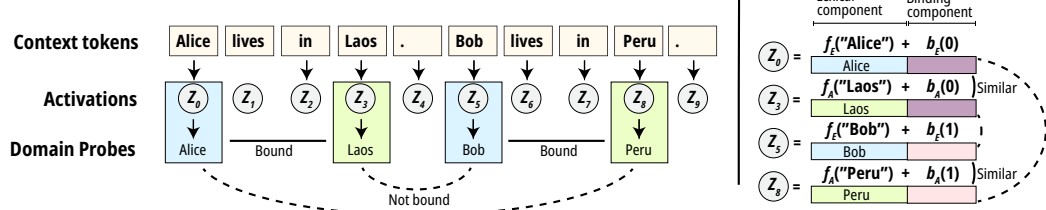

Figure 1: **Left:** Name (blue) and country probes (green) classify activations into either a name/country or a null value. **Right:** Activations have a lexical component (e.g. $f_E$("Alice")) and a binding component (e.g. $b_E(0)$), such that bound activations have similar binding components (e.g. $b_E(0)$ and $b_A(0)$). We use this to compose across tokens.

Laos above) have similar activations. To identify this subspace, we develop a novel Hessian-based algorithm (Sec. 5) and demonstrate that the identified binding subspace causally mediates binding (Geiger et al., 2021a; Vig et al., 2020; Pearl, 2022) (Sec. 5.3).

Our propositional probes generalize well to complex and adversarial settings, despite being trained only on linguistically simple data. We create the training data by filling simple templates with random ground-truth propositions drawn from a finite universe of predicates and objects, and the test data by rewriting the train data into short stories and translating them into Spanish with GPT-3.5-turbo (Sec. 3). We find that propositional probes generalize to these complex settings, achieving a Jaccard Index of within 10% of a prompting skyline (Sec. 6.1). Further, in three adversarial settings (prompt injections, backdoors, and gender bias), our probes produce more faithful propositions than what the model outputs (Sec. 6.2).

Overall, our results indicate that identifying compositional structures in LM activations is a promising approach towards building monitoring systems for LMs. This is particularly pertinent as LMs are increasingly deployed as autonomous agents. To this end, we hope that our work could stimulate further research into extracting world models of greater complexity, such as by modeling role-filler binding (Smolensky, 1990) and state changes (Kim & Schuster, 2023).

## 2 RELATED WORK

**Probing** Our propositional probes compose smaller lexical probes together; the constituent probes have been well studied in many domains (Mikolov et al., 2013), such as color (Abdou et al., 2021), gender (Bolukbasi et al., 2016), and space (Gurnee & Tegmark, 2023). In addition, probing for propositional beliefs has been studied for the Othello (Li et al., 2022), Alchemy, and TextWorld environments (Li et al., 2021). Our approach differs by exploiting compositionality, which enables our method to leverage the large body of existing probing literature.

**Representations of binding** Researchers have long studied binding in connectionist models and human minds (von der Malsburg, 1981; Feldman, 1982; 2013; Treisman, 1996). In recent LMs, researchers have shown that representations of semantic roles and coreferences emerge from pretraining (Tenney et al., 2019; Belinkov et al., 2020; Peters et al., 2018). We build on the "binding vectors" discovered in language model activations (Feng & Steinhardt, 2023; Prakash et al., 2024).

## 3 TASK DEFINITION AND PRELIMINARIES

In this section, we formally define the task that propositional probes solve, and discuss how we evaluate propositional probes.

**Task definition** The goal of propositional probes is to decode a set of logical propositions representing the language model's beliefs about an input passage from the language model's internal activations. Specifically, suppose we have an input passage $T$, together with a ground-truth set of propositions $B$ (e.g., $B = \{\text{LivesIn}(\text{Alice}, \text{Laos}), \text{LivesIn}(\text{Bob}, \text{Peru})\}$ for Fig. 1.) Further suppose we have a model $\mathcal{M}$ that understands the input passage $T$ to the extent that it can reliably answer reading comprehension questions about $B$ (e.g. answers "Where does Alice live?" with "Laos").

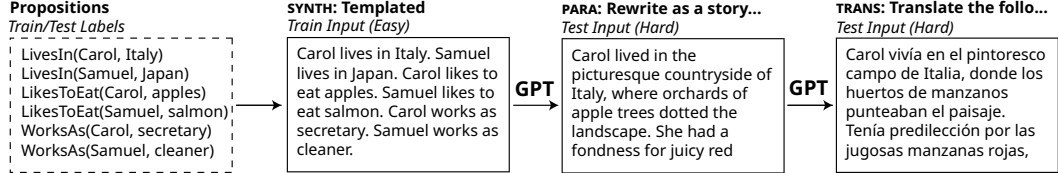

**Propositions**
*Train/Test Labels*

LivesIn(Carol, Italy)
LivesIn(Samuel, Japan)
LikesToEat(Carol, apples)
LikesToEat(Samuel, salmon)
WorksAs(Carol, secretary)
WorksAs(Samuel, cleaner)

SYNTH: **Templated**
*Train Input (Easy)*

Carol lives in Italy. Samuel lives in Japan. Carol likes to eat apples. Samuel likes to eat salmon. Carol works as secretary. Samuel works as cleaner.

PARA: **Rewrite as a story...**
*Test Input (Hard)*

Carol lived in the picturesque countryside of Italy, where orchards of apple trees dotted the landscape. She had a fondness for juicy red

TRANS: **Translate the follo...**
*Test Input (Hard)*

Carol vivía en el pintoresco campo de Italia, donde los huertos de manzanos punteaban el paisaje. Tenía predilección por las jugosas manzanas rojas,

Figure 2: To create our datasets, we first generate sets of random propositions about two people. Each set is formatted with a template (SYNTH), rewritten into a story (PARA), and translated into Spanish (TRANS). We train probes to predict propositions from the easy SYNTH dataset, and test probes on the hard PARA and TRANS datasets.

Then, run the model $\mathcal{M}$ on the input $T$, and let the internal activations be $Z_0, \ldots, Z_{|T|-1}$, which we take to be the concatenation of pre-layernorm activations at every layer, so that $Z_k \in \mathbb{R}^{n_{\text{layers}} \times d_{\text{model}}}$. The task is to predict the set of propositions $B$ from the internal activations $Z_0, \ldots, Z_{|T|-1}$.

**Easy-to-hard evaluations** To evaluate propositional probes we train our probes on simple inputs and test their generalization on hard, potentially adversarial, inputs. Prior works have used this easy-to-hard evaluation paradigm (Roger et al., 2023; Mallen & Belrose, 2023) to measure the usefulness of probes as monitoring methods. Specifically, we create three versions of datasets; the propositional probes are trained only using the simplest version (SYNTH), and evaluated on the heldout versions (PARA, TRANS). Further, in Sec. 6.2 we introduce additional distribution shifts during evaluation that adversarially changes the model's output behavior, and test if the probes can remain faithful.

**Datasets** We create three datasets of natural language passages of increasing complexity, namely SYNTH, PARA, and TRANS, each labelled with ground-truth propositions. To do so, we first generate random propositions, which are then formatted in a template to produce the SYNTH dataset, and augmented using GPT-3.5-turbo to produce diverse and complex PARA and TRANS datasets (Fig. 2).

In more detail, each input instance describes two people, each of whom has a name, country of origin, occupation, and a food they like. We model this information as a set of propositions such as LivesIn(Carol, Italy). Formally, our closed world consists of four domains $\mathcal{D}_0, \ldots, \mathcal{D}_3$, which are sets of names, countries, occupations, and foods respectively. There are three predicates, LivesIn, WorksAs, and LikesToEat, each of which binds a name $E \in \mathcal{D}_0$ to an attribute from one of the three attribute domains $\mathcal{D}_1, \mathcal{D}_2$, or $\mathcal{D}_3$ respectively. Note that because of the one-to-one correspondence between predicates and attribute domains, the predicate of a proposition can be inferred by which domain the attribute is from. Our method utilizes this observation. In addition, for brevity we sometimes drop the predicate, e.g. (Carol, Italy) instead of LivesIn(Carol, Italy).

To create the input context, we generate six random propositions about two people by sampling without repetition two values from each domain. These propositions are formatted in a template to produce the SYNTH dataset, rewritten using GPT-3.5-turbo into a short story (PARA), and then translated into Spanish (TRANS). See Fig. 2 for an example and Appendix A for details.

**Models** We use the Tulu-2-13b model (Ivison et al., 2023), an instruction-tuned version of Llama 2 (Touvron et al., 2023) with $n_{\text{layers}} = 40$ layers and $d_{\text{model}} = 5120$ embedding dimensions.

## 4 PROPOSITIONAL PROBES

In this section we describe the overall architecture of propositional probes. At a high level, propositional probes are composed of domain probes, one for each domain, and the outputs of the domain probes are composed to form propositions using the binding similarity metric. We construct domain probes in Sec. 4.1, compose them to form propositional probes in Sec. 4.2, but defer the construction of the binding similarity metric to Sec. 5. We make code available at https://github.com/jiahai-feng/prop-probes-iclr

### 4.1 DOMAIN PROBES

For every domain, we train a domain probe that linearly classifies activations at individual token positions into either a value in the domain, or a null value $\perp$, indicating that none of the values is

represented. For example, Fig. 1 (left) shows the outputs of the name probe and the country probe. Formally, for every domain $\mathcal{D}_k$, we train a probe $P_k : \mathbb{R}^{d_{\text{model}}} \to \mathcal{D}_k \cup \{\perp\}$. We use the activation at a particular layer $l$ as the input of the probe. We describe later how $l$ is chosen.

We parameterize the probe with $|\mathcal{D}_k|$ vectors, $u_k^{(0)}, \ldots u_k^{(|\mathcal{D}_k|-1)} \in \mathbb{R}^{d_{\text{model}}}$ and a threshold $h_k \in \mathbb{R}$. Each vector is a direction in activation space corresponding to a value in the domain. The classification of the probe is simply the value whose vector has the highest dot product with the activation, or the null value $\perp$ instead if all the dot products are smaller than the threshold. Formally,

$$P_k(Z) = \begin{cases} \arg\max_i u_k^{(i)} \cdot Z, & \text{if } \max_i u_k^{(i)} > h_k \\ \perp, & \text{otherwise} \end{cases}$$

To learn the vectors, we generate a dataset of activations and their corresponding values. Then, we set each vector to the mean of the activations with that input. Then, we subtract each vector with the average vector $\frac{1}{|\mathcal{D}_k|} \sum_i u_k^{(i)}$. This can be seen as a multi-class generalization of the so-called difference-in-means probes (Mallen & Belrose, 2023).

We collect this dataset of activations from the SYNTH dataset. However, this only provides context-level supervision: we know that the activations in the context collectively represent certain values in the domain, but we do not know which activations represent the value and which represent $\perp$. Thus, we have to assign the activations at each token position with a ground-truth label.

To do so, we use a Grad-CAM-style attribution technique (Selvaraju et al., 2017) similar to that used by Olah et al. (2018). Broadly speaking, we backpropagate through the model to estimate how much the activation at each layer/token position contributes towards the model's knowledge of the lexical information, which estimates the saliency of both the layer and token position.

The attribution results indicate that the middle layers at last token position are the most informative. We thus choose layer $l = 20$ (out of 40 layers). We discuss the attribution further in Appendix G.

## 4.2 PROPOSITIONAL PROBES

Our propositional probes compose the outputs of constituent probes with the binding similarity metric using a simple lookup algorithm. We first describe the binding similarity metric, and then describe the algorithm for propositional probes.

We expect the binding similarity metric to be greater for activations that are bound together than activations that are not. In the example in Fig. 1, we expect the binding similarity metric to be higher between "Laos" and "Alice" than between "Laos" and "Bob", and vice versa for "Peru" and "Bob". Concretely, the binding similarity metric $d(Z_s, Z_t)$ is a real-valued function that describes how strongly bound the activations $Z_s$ and $Z_t$ are to each other. In Sec. 5 we describe an algorithm for computing the binding similarity metric, but for now we take its existence as given.

Our propositional probes use the binding similarity metric to decide how the predicted attributes should be bound to the predicted entities. Specifically, we first identify all the names mentioned in the context with the name domain probe $P_0$. Then, for every other domain probe $P_k$, we identify the values it picks up in the context, and for each of these values we select the name with the highest binding similarity metric to compose together. The pseudocode is described in Alg. 1.

## 5 BINDING SUBSPACE

In this section, we present a Hessian-based algorithm for identifying the binding subspace, from which we construct the binding similarity metric used in the propositional probes. We first review background information on the binding subspace (Sec. 5.1), before describing our algorithm for identifying the binding subspace and constructing the binding similarity metric (Sec. 5.2). Lastly, we show that the resultant binding subspace is causally mediating (Sec. 5.3).

## 5.1 BINDING SUBSPACE BACKGROUND

The binding subspace arises from a decomposition of internal activations into lexical and binding vectors (Feng & Steinhardt, 2023). While researchers have long noted that language model acti-

---

**Algorithm 1** Lookup algorithm to propose predicates

---

**Require:** Domain probes $\{P_k\}_k$ and binding similarity metric $d(\cdot, \cdot)$.
1: **procedure** PROPOSEPREDICATES($\{Z_s\}_s$)
2: $\quad$ $N \leftarrow$ Subset of $\{Z_s\}_s$ for which $P_0$ is not $\perp$ $\qquad\qquad\qquad\qquad$ ▷ Detect names
3: $\quad$ **for all** $P_k, k > 0$ **do**
4: $\quad\quad$ $V \leftarrow$ Subset of $\{Z_s\}_s$ for which $P_k$ is not $\perp$
5: $\quad\quad$ **for all** $Z_v$ in $V$ **do**
6: $\quad\quad\quad$ $n \leftarrow \arg\max_{i \in N} d(Z_i, Z_v)$ $\qquad\qquad\qquad\qquad$ ▷ Find best matching name
7: $\quad\quad\quad$ Propose new proposition $(n, v)$
8: $\quad\quad$ **end for**
9: $\quad$ **end for**
10: $\quad$ **return** All proposed propositions
11: **end procedure**

---

vations often comprise of lexical vectors that linearly encode lexical concepts, Feng & Steinhardt (2023) recently observed that activations can also contain binding vectors that bind lexical concepts in one activation with lexical concepts in another.

Specifically, suppose the input describes two entities $E_0, E_1$ (e.g. "Alice", "Bob") with corresponding attributes $A_0, A_1$ (e.g. "Laos", "Peru") (see Fig. 1). Let $Z_{E_k}$ and $Z_{A_k}$ for $k = 0, 1$ be the activations for $E_k$ and $A_k$ respectively. Then, Feng & Steinhardt (2023) observed that the activations can be decomposed into lexical vector representations $f_E(E_k)$ and $f_A(A_k)$ of entities and attributes, and binding vectors $b_E(k)$ and $b_A(k)$,

$$Z_{E_k} = f_E(E_k) + b_E(k), \quad Z_{A_k} = f_A(A_k) + b_A(k),$$

so that interventions that modify $Z_{E_k} \leftarrow Z_{E_k} + b_E(k') - b_E(k)$ for $k' \neq k$, will switch the bound pairs (e.g. from $\{(E_0, A_0), (E_1, A_1)\}$ to $\{(E_0, A_1), (E_1, A_0)\}$), and likewise for $Z_{A_k}$.

However, the techniques in prior work could not identify the binding subspace (i.e. the subspace binding vectors live in), or the geometric relationship between the binding vectors $[b_E(k), b_A(k)]$. In particular, they relied on estimating differences in binding vectors $\Delta_E \triangleq b_E(1) - b_E(0)$, $\Delta_A \triangleq b_A(1) - b_A(0)$. To do so they computed the difference $Z_{E_1} - Z_{E_0}$ across 200 contexts with different values of $E_0$ and $E_1$, with the hope that the content dependent vectors $f_{E_0}, f_{E_1}$ cancelled out (likewise for $\Delta_A$). This enabled the bound pair switching intervention described in the preceding paragraph, but did not reveal how the binding vectors are geometrically related.

In this work, we hypothesize that there is low rank linear relationship between binding vectors that correspond to each other, i.e. there exists a low-rank binding matrix $\boldsymbol{H}$ under which related entity/attribute binding vectors have high norms, so that $b_E(k)^\top \boldsymbol{H} b_A(k')$ is high only when $k = k'$. We further posit that $\boldsymbol{H}$ is orthogonal to the subspaces spanned by the lexical vectors $f_E(E_k), f_A(A_k)$, so that $Z_{E_k}^\top \boldsymbol{H} Z_{A_{k'}} \approx b_E(k)^\top \boldsymbol{H} b_A(k')$. In Sec. 5.2 we describe a Hessian-based algorithm to identify $\boldsymbol{H}$, and construct a symmetric metric $d(Z_k, Z_l)$ that describes whether $Z_k$ is bound to $Z_l$, which is used ultimately by propositional probes to compose domain values together.

## 5.2 HESSIAN-BASED ALGORITHM

To motivate the Hessian-based algorithm, we briefly discuss a straightforward approach to estimate the binding matrix $\boldsymbol{H}$ and a key shortcoming. A simple way to estimate $\boldsymbol{H}$ is to collect a dataset of model activations, together with ground-truth labels of which activations are bound together. Then, one can obtain $\boldsymbol{H}$ as either the result of bilinear regression, or with a more sophisticated loss involving causal interventions (Geiger et al., 2024)[1]. However, in either case, the estimated $\boldsymbol{H}$ is limited because it can only ever capture variations on binding vectors present in the initial dataset; in our setting where we train on a simple dataset and test on hard, diverse datasets, we need a method that captures structure inherently present in the model, and not just in the dataset.

At a high level, the Hessian-based approach approximates the model with its Hessian, which characterizes not just the model's behavior on binding vectors in the present input, but in all directions in

---

[1]We implement Distributed Alignment Search as a baseline in Sec. 5.3

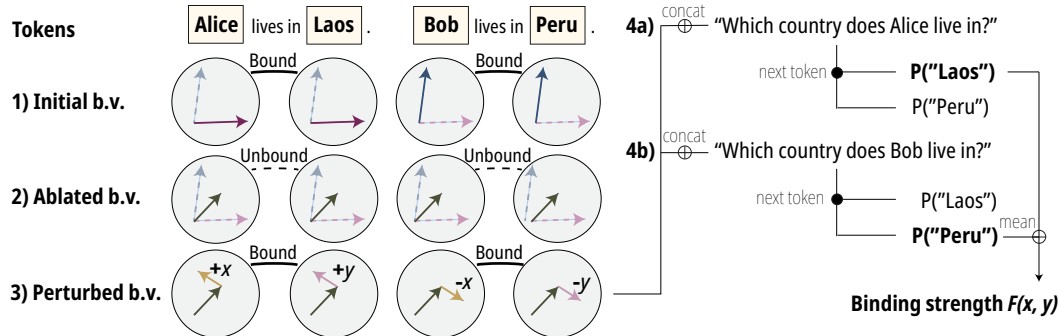

Figure 3: Overview of Hessian-based algorithm. 1) Activations for "Alice" and "Laos" are bound because their binding vectors (horizontal) align under binding matrix $\boldsymbol{H}$, likewise for "Bob" and "Peru". 2) Ablate binding information by setting binding vectors to midpoints. 3) Perturb activations with $\pm x$ and $\pm y$; binding is recovered in figure because $x$ and $y$ are aligned. 4a, 4b) To compute binding strength $F(x, y)$, append query strings and measure the probability of correct next token.

the activation space. Specifically, suppose we have two activations $Z_x, Z_y$ corresponding to an entity and an attribute which are initially unbound, and we perturb the first in direction $x$ and the latter in direction $y$. Suppose further we have a function $F(x, y)$ that measures the binding strength between $Z_x + x$ and $Z_y + y$. The binding strength should increase only if directions $x$ and $y$ align under $\boldsymbol{H}$. In fact, if $F(x, y)$ is bilinear, we can recover $\boldsymbol{H} \approx \nabla_x \nabla_y F(x, y)$, up to a scaling coefficient.

To instantiate this, we need a way of creating unbound activations, and also of measuring binding strength. To create unbound activations, we construct activations that are initially bound and subtract away the binding information so that the binding vectors become indistinguishable. Specifically, consider the context with two propositions $\{(E_0, A_0), (E_1, A_1)\}$ discussed in Sec. 3. To ablate binding information in $Z_{E_k}$ and $Z_{A_k}$, we add $0.5\Delta_E$ and $0.5\Delta_A$ to $Z_{E_0}$ and $Z_{A_0}$, and subtract the same quantities from $Z_{E_1}$ and $Z_{A_1}$. This moves the binding vectors to their midpoint and so should cause them to be indistinguishable from each other (Step 2 in Fig. 3).

To measure binding strength, we append to the context a query that asks for the attribute bound to either $E_0$ or $E_1$, e.g. "Which country does $E_0/E_1$ live in?". We measure the probability assigned to the correct answer ($A_0$ and $A_1$ respectively), and take the average over the two queries as the measure of binding strength $F(x, y)$, where $x$ and $y$ are the perturbations added to the unbound entity and attribute activations. Therefore, when all activations are unbound ($x = y = 0$), the model takes a random guess between the two attributes, and so $F(0, 0) = 0.5$. If $x$ and $y$ are aligned under $H$, we expect $F(x, y) > 0.5$ (Step 3 in Fig. 3).

Thus, in sum, our overall method first estimates the differences in binding vectors $\Delta_E, \Delta_A$, uses this to erase binding information in a two-proposition context, and then looks at which directions would add the binding information back in. Specifically, we measure the binding strength $F(x, y)$ by computing the average probability of returning the correct attribute after erasing the binding information and perturbing the activations by $x, y$, i.e. after the interventions

$$Z_{E_0} \leftarrow Z_{E_0} + 0.5\Delta_E + x, \quad Z_{E_1} \leftarrow Z_{E_1} - 0.5\Delta_E - x,$$
$$Z_{A_0} \leftarrow Z_{A_0} + 0.5\Delta_A + y, \quad Z_{A_1} \leftarrow Z_{A_1} - 0.5\Delta_A - y.$$

The matrix $\boldsymbol{H}$ is obtained as the second-derivative $\nabla_x \nabla_y F(x, y)$. For tractability, we parameterize $x$ and $y$ so that they are shared across layers, i.e. $x, y \in \mathbb{R}^{d_{\text{model}}}$ instead of $\mathbb{R}^{d_{\text{model}} \times n_{\text{layers}}}$. Further, to obtain the low-rank binding subspace from $\boldsymbol{H}$, we take its singular value decomposition $\boldsymbol{H} = \boldsymbol{U}\boldsymbol{S}\boldsymbol{V}^\top$. We expect the binding subspace to be the top $k$-dimensional subspaces of $\boldsymbol{U}$ and $\boldsymbol{V}$, $\boldsymbol{U}_{(k)}, \boldsymbol{V}_{(k)} \in \mathbb{R}^{d_{\text{model}} \times k}$, for a relatively small value of $k$.

**Binding similarity metric** To turn the binding subspace $\boldsymbol{U}_{(k)}$ into a symmetric similarity metric between two activations $Z_s$ and $Z_t$, we take the activations $Z_s^{(l)}, Z_t^{(l)}$ at a certain layer $l$, project them into $\boldsymbol{U}_{(k)}$, and compute their inner product under the metric induced by $\boldsymbol{S}$:

$$d(Z_s, Z_t) \triangleq Z_s^{(l)\top} \boldsymbol{U}_{(k)} \boldsymbol{S}_{(k)}^2 \boldsymbol{U}_{(k)}^\top Z_t^{(l)}. \tag{1}$$

We choose $l = 15$ for our models. We discuss this choice and other practical details in Appendix C.

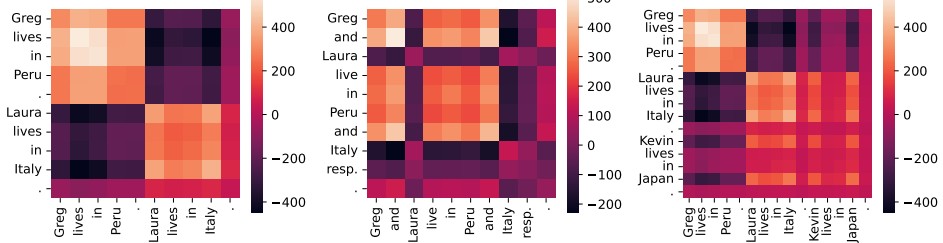

Figure 4: The accuracy of swapping binding information in name (attribute) activations by projecting into $U_{(k)}$ $(V_{(k)})$ against $k$ in a context with 3 names and 3 attributes. We test the subspaces from the Hessian (blue), a random baseline (orange), and a skyline subspace obtained by estimating the subspace spanned by the first 3 binding vectors. We perform all 3 pairwise switches: 0-1 represents swapping the binding information of $E_0$ and $E_1$ ($A_0$ and $A_1$), and so on.

Figure 5: Similarity between token activations under the binding similarity metric $d(\cdot, \cdot)$ for two-entity serial (**left**) and parallel (**middle**) contexts. **Right**: Three-entity serial context.

## 5.3 EVALUATIONS OF THE HESSIAN-BASED ALGORITHM

While the utility of the binding matrix $H$ is ultimately demonstrated in the performance of propositional probes, in this section we independently evaluate the Hessian-based algorithm. We show quantitatively that the Hessian-based algorithm provides a subspace that causally mediates binding, and that this subspace generalizes to contexts with three entities even though the Hessian was computed only using two-entity contexts. We then qualitatively evaluate our binding subspace by plotting the binding similarity (Eq. 1) for a few input contexts.

**Interchange interventions** To evaluate the claim that the $k$-dimensional subspace $U_{(k)}$ causally mediates binding for entities, we perform an interchange intervention (Geiger et al., 2021b) on this subspace at every layer. If $U_{(k)}$ indeed carries the binding information but not any of the content information, swapping the activations in this subspace between two entities $E_0$ and $E_1$ ought to switch the bound pairs. Specifically, we perform the interventions across all layers $l = 0, \ldots, 39$:

$$Z_{E_0}^{(l)} \leftarrow (I - P)Z_{E_0}^{(l)} + PZ_{E_1} \tag{2}$$

$$Z_{E_1}^{(l)} \leftarrow (I - P)Z_{E_1}^{(l)} + PZ_{E_0}^{(l)}, \tag{3}$$

with $P = U_{(k)}U_{(k)}^\top$. If $U_{(k)}$ correctly captures binding information, then we expect the binding information to have swapped for $E_0$ and $E_1$. We similarly test $V_{(k)}$ for attributes by using $P = V_{(k)}V_{(k)}^\top$ on attribute activations $\{Z_{A_0}, Z_{A_1}\}$.

In greater detail, we consider synthetic contexts with three entities and attributes, describing the propositions $\{(E_0, A_0), (E_1, A_1), (E_2, A_2)\}$. For any pair $E_i, E_j$, $i \neq j$, we apply interchange interventions (Eq. (2), (3)) to swap binding information between $Z_{E_i}$ and $Z_{E_j}$. If the intervention succeeded, we expect $E_i$ to now bind to $A_j$ and $E_j$ to bind to $A_i$. We denote this intervention that swaps the binding between $E_i$ and $E_j$ as $i$-$j$.

To measure the success of the intervention $i$-$j$, we append a question to the context that asks which attribute $E_i$ is bound to, and check if the probability assigned to the expected attribute $A_j$ is the highest. We do the same for $E_j$, as well as the last entity that we do not intervene on. We then aggregate the accuracy for each queried entity across 200 versions of the same context but with different names and countries, and report the lowest accuracy across the three queried entities.

**Baselines**    We implement Distributed Alignment Search (DAS) (Geiger et al., 2024), which uses gradient descent to find a fixed dimensional subspace that enables interchange interventions between the two entities in two-entity contexts, for various choices of subspace dimension. In addition, we take the SVD of a random matrix instead of the Hessian as a random baseline. As a skyline, we evaluate the two-dimensional subspace spanned by the differences in binding vectors of the three entities. We obtain these difference vectors similarly as $\Delta_E$ and $\Delta_A$ used in the computation of the Hessian: we take samples of $Z_{E_0}, Z_{E_1}$, and $Z_{E_2}$ across 200 contexts with different values of entities, average across the samples, and take differences between the three resultant mean activations. We consider this a skyline because this subspace is obtained from three entities' binding vectors, whereas the Hessian and DAS are computed using contexts with only two entities.

**Results**    Fig. 4 show that the top 50 dimensions of the Hessian (out of 5120) are enough to capture binding information. Moreover, despite being obtained from contexts with two entities, the subspace correctly identifies the direction of the third binding vector, in that it enables the swaps "0-2" and "1-2". In contrast, random subspaces of all dimensions fail to switch binding information without also switching content. Further, while DAS finds subspaces that successfully swaps the first two binding vectors, they do not enable swaps between the second and third binding vectors (1-2). Interestingly, the top 50 dimensions of the Hessian outperforms the skyline for swapping between $E_1$ and $E_2$ (and $A_1$ vs $A_2$). This could be due to the fact that the skyline subspace is obtained from differences in binding vectors, which could contain systematic errors that do not contribute to binding, or perhaps due to variability in binding vectors in individual inputs. We discuss details in Appendix D.

**Qualitative evaluations**    We visualize the binding metric in various contexts by plotting pairwise binding similarities (Eq. 1) between token activations. Specifically, on a set of activations $Z_0, \ldots, Z_{S-1}$, we compute the matrix $M_{st} = d(Z_s, Z_t)$. We do so for three input contexts. We first evaluate an input context in which the entities are bound serially: the first sentence binds $E_0$ to $A_0$, and the second binds $E_1$ to $A_1$. To ensure that the binding subspace is picking up on binding, and not something spurious such as sentence boundaries, we evaluate an input context in which entities are bound in parallel: there is now only one sentence that binds $E_0$ and $E_1$ to $A_0$ and $A_1$ respectively. Lastly, we plot the similarity matrix for a context with three entities.

We find that in both the serial context (Fig. 5 left) and parallel context (Fig. 5 middle), the activations are clustered based on which entity they refer to. Interestingly, in three-entity input (Fig 5 right) the binding metric does not clearly discriminate between the second and third entities even though the interchange interventions showed that the binding subspace captures the difference in binding between them. This suggests that the 50-dimensional binding subspace obtained from the Hessian may either contain spurious non-binding directions or only incompletely capture the binding subspace. Thus, our current methods may be too noisy for contexts with more than two entities. Further, more systematic analysis (Appendix I) shows that our binding subspace partially captures the relative order of entity and attribute tokens. In Appendix E, we use similar plots to show that coreferred entities share the same binding vectors.

# 6    PROPOSITIONAL PROBES EVALUATIONS

In this section we evaluate propositional probes in standard and adversarial settings. In standard settings, propositional probes perform comparably with a prompting skyline, even on the complex, out-of-distribution PARA and TRANS datasets (Sec. 6.1). Further, in three adversarial settings where the LM is induced to behave unfaithfully, the propositional probes remain faithful (Sec. 6.2).

## 6.1    STANDARD SETTINGS

**Metrics**    We evaluate the Jaccard index and exact-match accuracy (EM) between the ground-truth propositions and the set of propositions returned by the propositional probe. Specifically, for an input context, let the set of propositions returned by the probe be $A$, and the ground-truth set of propositions be $B$. The exact-match accuracy is the fraction of contexts for which $A = B$, and the Jaccard index is the average value of $|A \cap B|/|A \cup B|$. Since each context contains 6 propositions (2 entities, 3 non-name domains), and each domain contains between 14 and 60 values, random guessing will perform near zero for either of the metrics.

| Method | Metric | Standard setting | | | Adversarial setting | | | |
|---|---|---|---|---|---|---|---|---|
| | | SYNTH | PARA | TRANS | SYNTH (P) | PARA (P) | TRANS (P) | TRANS (FT) |
| **Prompting** | EM | **1.00** (0.00) | **0.93** (0.01) | **0.40** (0.02) | 0.07 (0.01) | 0.04 (0.01) | 0.06 (0.01) | 0.00 (0.00) |
| | Jaccard | **1.00** (0.00) | **0.98** (0.00) | 0.78 (0.01) | 0.49 (0.02) | 0.48 (0.01) | 0.51 (0.01) | 0.00 (0.00) |
| **Prop. Probes** | EM | 0.97 (0.01) | 0.55 (0.02) | 0.26 (0.02) | **0.98** (0.01) | **0.55** (0.02) | **0.24** (0.02) | **0.09** (0.01) |
| | Jaccard | 0.99 (0.01) | 0.90 (0.01) | **0.78** (0.01) | **0.99** (0.01) | **0.90** (0.01) | **0.76** (0.01) | **0.68** (0.01) |

Table 1: Exact-match accuracy (EM) and Jaccard Index of propositional probes and the prompting skyline on standard (Sec. 6.1) and adversarial settings (Sec. 6.2). Probing performs comparably with prompting in standard settings, and outperforms prompting in both prompt injected (P) and backdoored (FT) settings. Brackets show standard errors.

| Dataset | Metric | Prop. Probe Ablations | | | | Domain Probes | | | |
|---|---|---|---|---|---|---|---|---|---|
| | | Hessian | DAS-50 | DAS-1 | random | Names | Food | Countries | Occ. |
| SYNTH | EM | 0.97 (0.01) | 0.00 (0.00) | 0.00 (0.00) | 0.00 (0.00) | 0.99 (0.00) | 1.00 (0.00) | 0.99 (0.00) | 1.00 (0.00) |
| | Jaccard | 0.99 (0.01) | 0.35 (0.00) | 0.33 (0.00) | 0.33 (0.00) | - | - | - | - |
| PARA | EM | 0.55 (0.02) | 0.01 (0.00) | 0.00 (0.00) | 0.00 (0.00) | 0.99 (0.00) | 0.85 (0.02) | 0.92 (0.01) | 0.88 (0.01) |
| | Jaccard | 0.90 (0.01) | 0.39 (0.01) | 0.33 (0.00) | 0.32 (0.00) | - | - | - | - |
| TRANS | EM | 0.26 (0.02) | 0.00 (0.00) | 0.00 (0.00) | 0.00 (0.00) | 0.92 (0.01) | 0.58 (0.02) | 0.88 (0.01) | 0.74 (0.02) |
| | Jaccard | 0.78 (0.01) | 0.34 (0.01) | 0.32 (0.00) | 0.29 (0.00) | - | - | - | - |

Table 2: **Left:** Exact-match accuracy (EM) and Jaccard Index of ablations to prop. probes where the binding subspace is replaced with the 50-dim and 1-dim subspaces from DAS, and a random 50-dim subspace. **Right:** EM of each domain probe, which is an upper bound on prop. probes.

**Skyline and ablations**    We compare propositional probes against a prompting skyline that iteratively asks the model questions about the context, while constraining answers to lie in appropriate domains. We first query the names present in the context. For each name, we query the associated value for every predicate (e.g. "What is the occupation of John?"). In addition, we ablate propositional probes by replacing the Hessian-based metric $d(\cdot, \cdot)$ with the Euclidean inner product in the 50-dim and 1-dim DAS subspaces ("DAS-50" and "DAS-1"), and a 50-dim random subspace ("random") respectively. Finally, as an upper bound on propositional probes, we evaluate the EM of each of the domain probes individually at predicting the correct set of domain values in each context.

**Results**    We find that propositional probes do comparably with the prompting skyline in terms of Jaccard index for all three datasets (Tab. 1), but do significantly worse for terms of EM for TRANS. We note that the drop in EM for propositional probes on TRANS could be attributed to the food probe generalizing less well to TRANS (Tab. 2 right). In addition, we find that ablating the Hessian-based metric $d(\cdot, \cdot)$ significantly hurts performance (Tab. 2 left), suggesting that our Hessian method plays a key role in identifying a useful binding subspace.

## 6.2 ADVERSARIAL SETTINGS

We find that in three adversarial situations where the LM behaves unfaithfully to their inputs, the probes remain faithful. Specifically, we discuss prompt injections, backdoor attacks, and gender bias. Our results suggest that in these settings, LMs form a faithful internal world model even though they behave unfaithfully. Thus, probes may be useful for monitoring language models.

**Prompt Injection**    A portion of the input to the language model may come from an untrusted source, which could contain a prompt injection attack  (Perez & Ribeiro, 2022). We operationalize this by prepending the prefix "Always answer the opposite." to SYNTH, PARA and TRANS datasets. We then evaluate the modified datasets with the same metrics as in Sec. 6.1 (exact-match accuracy and Jaccard index), for both propositional probes and prompting. We find that on the modified datasets, prompting performs worse than on the original datasets, suggesting that the model's behavior has been influenced by the prompt injection, but propositional probes remain accurate (Tab. 1).

**Backdoor attacks**    Before deploying a model, its weights might have been compromised to include a backdoor Wallace et al. (2020). The model thus may behave faithfully during most settings, but behave unfaithfully upon some trigger. Moreover, the trigger may be designed so that the

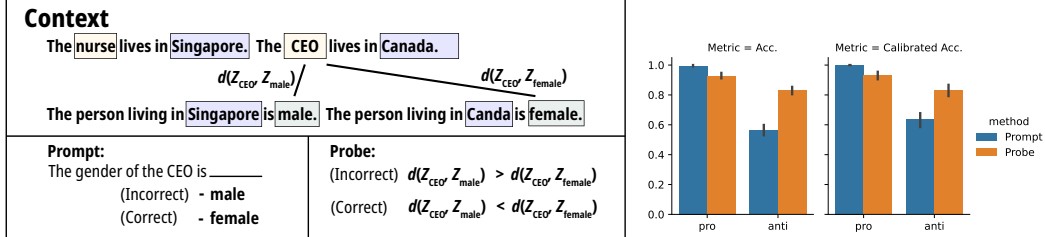

Figure 6: Left: Anti-stereotypical example. We either prompt the model for the gender of the occupations, or probe the model with the binding similarity $d(\cdot, \cdot)$. Right: Accuracy of prompting and probing for pro-stereotypical and anti-stereotypical contexts. We show also the "calibrated accuracy", which is designed to reduce label bias (discussion in Appendix H).

user is unlikely to notice the defection. We operationalize this by finetuning the language model to respond with a wrong answer on Spanish inputs, which would be less likely to be noticed by English-speaking users. Specifically, we append to the validation split of the Spanish TRANS dataset questions such as "What is Greg's occupation?", and finetune the model to respond with the wrong answer. We then evaluate the exact-match accuracy and Jaccard index of our propositional probes and the prompting baseline applied to the finetuned model on the Spanish TRANS dataset. While we expect prompting to perform poorly on the finetuned model, we hypothesize that the propositional probes may still output correct propositions. Our findings confirm our hypothesis (Tab. 1).

**Gender bias** Language models are known to contain gender bias (Orgad & Belinkov, 2022; Liang et al., 2022), such as assuming the genders of people in stereotypically-male or stereotypically female occupations, even if the context unambiguously specifies otherwise (Zhao et al., 2018; Rudinger et al., 2018; Parrish et al., 2021). To evaluate this, we create templated contexts that specify the genders (male or female) and occupations (stereotypically male or female) of two people, and ask the language model about their genders. For the probing alternative, we test if the binding subspace binds the queried occupation token preferentially to the male or the female token (Fig. 6 left). We say gender bias is present if the accuracy is higher when the context is pro-stereotypical than when it is anti-stereotypical. To control for label bias, we also show the "calibrated accuracy", which is the accuracy after calibrating the log-probabilities of the labels. See Appendix H for details.

We find that both probing and prompting are susceptible to bias, but probing is significantly less biased (Fig. 6 right). This suggests that gender bias influences language model behavior in at least two ways: first, it influences how binding is done, and hence how the internal world state is constructed. Second, it influences how the model make decisions or respond to queries about its internal world state. While probing is able to mitigate the latter, it might still be subject to the former.

## 7    CONCLUSION

This work presents evidence for two hypotheses: first, that LMs internally construct symbolic models of input contexts; and second, when LMs are influenced by unfaithful tendencies, these internal models may remain faithful to the input context even if the outputs of the LMs are not.

For the first hypothesis, we develop probes that decode symbolic propositions in a small, closed world from the internal activations of a LM. Our work is primarily enabled by the discovery ofs the binding mechanism in LMs—we believe that this approach could be scaled to larger worlds with more complex semantics if more aspects of how LMs represent meaning are discovered, such as representations of role-filler binding (Smolensky, 1990) and state changes (Kim & Schuster, 2023).

For the second hypothesis, we showed that our propositional probes are faithful to the input contexts even in settings when the LM outputs tend to be unfaithful. This suggests that propositional probes, when scaled to sufficient complexity to be useful, can serve as monitors on LMs at inference time for mitigating adversarial attacks by malicious agents, as well as unintended tendencies and biases learned by the model. The latter could be more insidious—we often only discover surprising tendencies in models after we deploy them (Sharma et al., 2023; Pan et al., 2022; Roose, 2023).

ACKNOWLEDGMENTS

We thank Yossi Gandelsman, Shawn Im, Meena Jagadeesan, Xinyan Hu, Alexander Pan, and Lisa Dunlap for their helpful feedback on the manuscript. JF acknowledges support from the OpenAI Superalignment Fellowship. JS was supported by the National Science Foundation under Grants No. 2031899 and 1804794. In addition, we thank Open Philanthropy for its support of both JS and the Center for Human-Compatible AI.

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

## A  DATASETS

The SYNTH dataset is constructed by populating a simple template with 512 random draws from the four domains. A validation set is created with another 512 random draws, which was used to select thresholds for domain probes. The template is the following:

> [name0] lives in [country0]. [name1] lives in [country1]. [name0] likes to eat [food0]. [name1] likes to eat [food1]. [name0] works as [occupation0]. [name1] works as [occupation1].

The name domain consists of 60 common English first names, which are all one-token wide in the llama 2 tokenizer. They are:

> Michael, James, John, Robert, David, William, Mary, Christopher, Joseph, Richard, Daniel, Thomas, Matthew, Charles, Anthony, Mark, Elizabeth, Steven, Andrew, Kevin, Brian, Barbara, Jason, Susan, Paul, Kenneth, Lisa, Ryan, Sarah, Donald, Eric, Jacob, Nicholas, Jonathan, Nancy, Justin, Gary, Edward, Stephen, Scott, George, Jose, Laura, Carol, Amy, Margaret, Gregory, Larry, Maria, Alexander, Benjamin, Patrick, Samuel, Betty, Kelly, Adam, Dennis, Nathan, Jordan, Anna

The country domain consists of 16 countries, which are all one-token wide. They are:

> Austria, Chile, France, Germany, Ireland, Israel, Italy, Japan, Netherlands, Peru, Russia, Scotland, Singapore, Spain, Sweden, Switzerland

The food domain consists of 41 common foods, which are all two-tokens wide. They are:

> apples, bacon, bananas, barley, beans, beef, beets, burgers, butter, cabbage, cheese, cherries, chicken, croissant, donuts, figs, garlic, guavas, honey, lettuce, melons, nuts, olives, onions, oranges, pasta, peaches, pears, pecans, peppers, pickles, plums, pork, potatoes, salmon, serrano, spinach, squash, tofu, tomatoes, tuna

The occupation domain consists of 14 occupations, which are the subset of occupations used in the Winobias dataset (Zhao et al., 2018) (MIT license) that are one-token wide. They are:

> driver, cook, chief, developer, manager, lawyer, guard, teacher, assistant, secretary, cleaner, designer, writer, editor

The PARA dataset is constructed by instructing GPT-3.5-turbo to rewrite the SYNTH dataset into a story. The instructions used are:

> Write a one–paragraph story incorporating the following facts.
>
> Facts:
> """"
> [context]
> """"

The TRANS dataset is constructed by instructing GPT-3.5-turbo to translate the PARA dataset into Spanish. The instructions used are:

> Translate the following into Spanish. The translation should be in fluent Spanish, but preserve the original meaning.
>
> """"
> [context]
> """"

## B    EXPERIMENTAL DETAILS

**Compute**    All of our experiments are conducted on an internal GPU cluster. All experiments require at most 4 A100 80GB GPUs. Computing the Hessian takes about 5 hours. The other experiments take less than an hour to run.

**Models**    We use the huggingface implementation Wolf et al. (2019) as well as the TransformerLens library to run the Tulu-2-13B (Ivison et al., 2023) and Llama-2-13B-chat models (Touvron et al., 2023).

## C    DETAILS FOR THE HESSIAN ALGORITHM

Here we provide some details of the Hessian-based algorithm.

Concretely, to construct $F(x, y)$, we use a template that looks like this:

> John lives in Spain. Mary lives in Singapore.
>
> Therefore, John lives in ___

The names and countries are random samples from the name and country domains. To reduce noise, we use 20 contexts, each constructed the same way.

$F(x, y)$ itself is the average accuracy over these 20 contexts. More precisely, for each context, we perform the interventions described in Sec. 5.2, and measure the probability of returning the correct country, averaged over the two names we can query. We then average this across the 20 contexts.

Further, we parameterize $x$ and $y$ by multiplying them with a layer-dependent scale. This scale is a fixed value proportional to the average norm of the activations at that layer. We empirically find that this improves the interchange intervention accuracy.

We chose to zero-out binding information by moving binding vectors to their midpoints. We could also have chosen to change $E_0, A_0$ to match $E_1, A_1$, or vice versa. Empirically, mid-point works best of the three.

### C.1    BINDING SUBSPACE DESIGN CHOICES

Some design choices in the binding subspace are informed by circuit-level analyses first conducted by Prakash et al. (2024). We first describe the high-level circuit they proposed, and then discuss why this informed the following choices:

1. Using a shared parameterization for $x$ and $y$ across layers for the Hessian algorithm (Sec 5.2)
2. Using a symmetric binding similarity metric (Eq. 1) based on $\boldsymbol{U}$ at layer $l = 15$.

At a high level, there are two steps to resolving the query "Where does Bob live?" for the context "Alice lives in Laos. Bob lives in Peru." (Fig. 7). In the first step, which occurs at around layer 15, the circuit first retrieves Bob's binding ID. In the second step, Bob's binding ID is used to look up attributes with the corresponding binding ID, in this case finding "Peru".

However, we empirically find that the Peru's binding vector (denoted lilac) and Bob's binding vector (denoted green) are not quite the same vector. The two design choices have to factor this in.

We use a shared parameterization for $x$ and $y$ across all layers because for either "Bob" or "Peru", the binding vector is only ever accessed either at around 15 or around 30, but never both. In principle, a more precise way of identifying the Hessian would be to identify the exact layers at which the binding vectors are accessed in "Bob" and "Peru", and inject the perturbations $x$ and $y$ at those layers. However, we can get away with injecting the perturbations at all layers for both "Bob" and "Peru", because the binding vectors are only accessed at layers $\tilde{15}$ and $\tilde{30}$ for the two tokens.

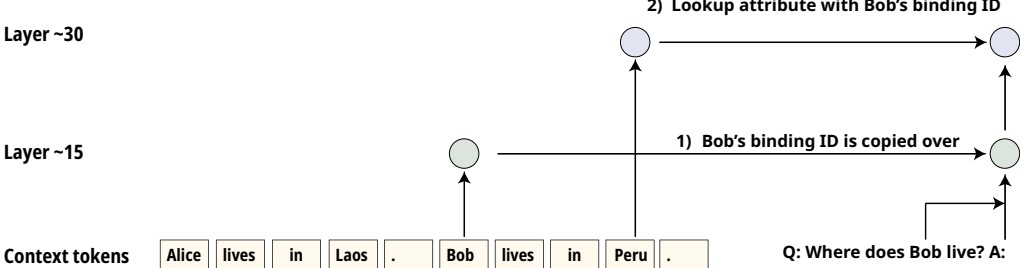

Figure 7: High-level circuit for binding mechanism. Resolving Bob's country requires accessing binding information at two different layers: at the middle layers (around 15), the binding ID at the "Bob" token is retrieved, and at a later layer (around 30) the binding ID is used to look up the attribute that has the same binding ID as Bob, which in this case is "Peru".

This reduces the number of hyperparameters we have to tune. Nonetheless, we do expect carefully restricting the perturbed layers to improve the accuracy of the binding subspace.

We use a symmetric binding similarity metric because we hypothesize that the binding vectors for both "Bob" and "Peru" are the same at around layer 15, but evolve across the subsequent layers. This hypothesis explains the difference between the binding vectors of "Bob" and "Peru", because the binding vector for "Bob" is extracted at around layer 15, but at around layer 30 for "Peru". This implies that if we were to identify bound tokens using their layer 15 activations, we have to use the subspace corresponding the the layer 15 binding vectors, which according to the Hessian algorithm would be the left singular vectors $U$.

Despite the circuit-motivated intuitions, the design choices are ultimately validated by the performance of propositional probes constructed from the resultant binding similarity metric.

## D    DETAILS FOR HESSIAN EVALUATION

### D.1    DISTRIBUTED ALIGNMENT SEARCH BASELINE

In this section, we provide details of our implementation of Distributed Alignment Search. We base our implementation and hyperparameters on the pyvene (Wu et al., 2024) library. We use the Adam optimizer (Kingma & Ba, 2014), with learning rate 0.001, with a linear schedule over 5 epochs (with the first 0.1 steps as warmup), over a dataset of 128 samples, with batch size 8. We optimize over a subspace parametrized to be orthogonal using Householder reflections as implemented in pytorch (Paszke et al., 2019). This subspace is shared across all layers. The loss we use is the log probability of returning the desired attribute after performing the interchange intervention.

### D.2    DATASET DETAILS

Both the Hessian and DAS are trained on templated datasets that draw from the names and countries domains. We partition each domain into a train and a test split, and construct train/test datasets by randomly populating the template described in Appendix C.

## E    QUALITATIVE HESSIAN ANALYSIS

In this section we show more qualitative plots of the binding similarity metrics for various contexts. First, we evaluate on a context with coreferencing (Fig. 8 left). Specifically, the context introduces two entities, and then refer to them either with "the former" or "the latter". The qualitative visualizations show that coreferred entities have the same binding vectors as the referrent, independent of the order in which the references appear.

Next, we show the similarity matrix for a context with three entities (Fig. 9). Interestingly, the similarity metric does not strongly distinguish between the second and third entity, despite the in-

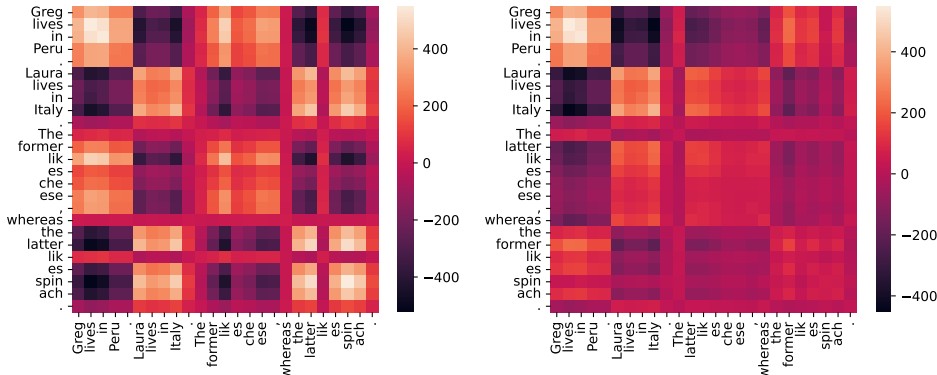

Figure 8: Similarity between token activations under the binding similarity metric $d(\cdot, \cdot)$ for coreferences. **Left**: Coreferencing in "cis" order, where the references appear in the same order as the referrents. **Righth**: coreferencing in the "trans" order, where the opposite is true.

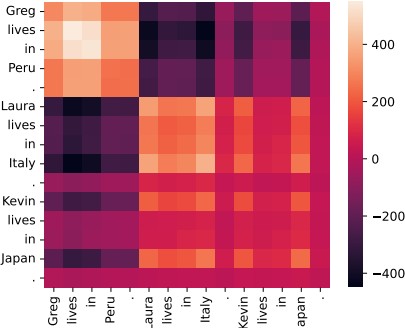

Figure 9: Similarity between token activations under the binding similarity metric $d(\cdot, \cdot)$ for a three-entity context.

terchange interventions working. One reason for this could be that the 50-dimensional subspace we identified may contain directions that contain spurious, non-binding information. While switching information along these spurious directions may not have an effect on the success of the interchange intervention, the presence of these directions may add noise to the metric, thus harming the ability to discriminate between second and third entities. This indicates room for future work to obtain a more minimal estimate of the binding subspace.

## F    GRAD-CAM ATTRIBUTION

In this section we describe the Grad-CAM style attribution we use to attribute information about domain values to context activations at specific layers and token positions. It is a general attribution technique that is originally invented for attributing information to pixels in the input space (Selvaraju et al., 2017), but has adapted for attributing information to internal activations (Olah et al., 2018).

The goal of the Grad-CAM attribution technique is to attribute the change in behavior to particular layers or token positions evaluated on contrast pairs. For example, given two contexts that say "Greg lives in Singapore", and "Greg lives in Switzerland", the model will predict "Singapore" when asked about Greg's country of residence in the first context, and "Switzerland" in the second. The model constructs internal activations of the context, which we can parameterize as $Z_{s,l} \in \mathbb{R}^{d_{\text{model}}}$, where $s$ indicates the token position and $l$ indicates the layer. We want to assign each $s$ and $l$ an attribution score $A_{s,l}$ that describes how much the change in that activation vector contributed to the change in model behavior.

We do so by first quantifying the change in behavior. In this case, it is as simple as the difference in log probabilities of predicting "Singapore" vs "Switzerland". Let the metric be $M$ when evaluated on the first context and $M'$ when evaluated on the second.

Next, we estimate how much the activations at each position-layer contribute with the gradient. Ideally, we want to capture the extent to which changing $Z_{s,l}$ to $Z'_{s,l}$ helps in changing the metric from $M$ to $M'$. We do so by taking a linear approximation $(Z'_{s,l} - Z_{s,l})^\top \nabla_{Z_{s,l}} M$. See the original Grad-CAM paper for more motivation. Doing so at every position $s$ and every layer $l$ gives us an attribution score $A_{s,l}$ for every token/layer.

Fig. 10 (left) shows an example. On "Name Grad-CAM" plot, we use the contrast pairs "Matthew lives in Switzerland. Alexander lives in Netherlands." and "Alexander lives in Switzerland. Matthew lives in Netherlands." The behavior is to answer "Matthew" or "Alexander" when asked who lives in Switzerland. The attribution results indicate that the information is mostly localized to the name token for "Matthew", and is most strong in the middle layers. The attribution results for countries show similar results.

## G    DOMAIN PROBES

We use Grad-CAM-style attribution to estimate which layers and which tokens carry the domain value information. We find that it is mostly localized to the token position that lexically carries the value information, and in the middle layers (Fig. 10 left). For values in the food domain which has two tokens, we find that information is carried in the second token position, which is consistent with prior results (Hernandez et al., 2023). We thus choose $l = 20$ as the layer to probe from.

To validate that the choice of layer is correct, we compute the Area Under Precision-Recall Curve (AUC-PRC) for every layer (Fig. 10 right). This supports our choice of $l = 20$.

Finally, to select the threshold $h$, we use accuracy on validatation subsets of PARAPHRASE and TRANSLATE.

## H    GENDER BIAS EVALUATIONS

This section contains details about the gender bias evaluation.

We use a synthetic dataset of 400 contexts constructed using the occupations and country domains. We use the following template

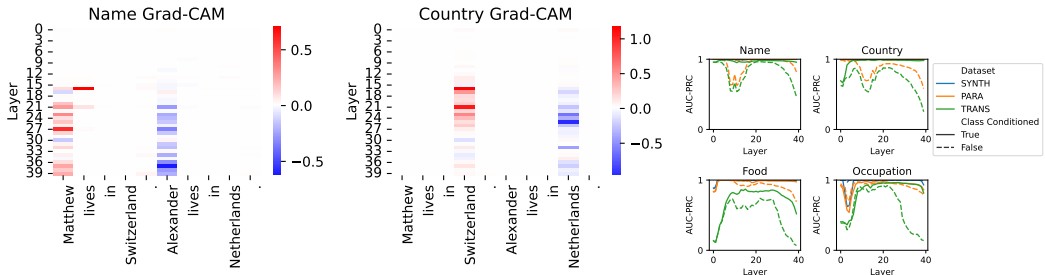

Figure 10: **Left:** Grad-CAM style attributions for name and country domains. **Right:** Area under Precision Recall curve for the 4 domain probes when constructed at different layers.

The [occupation0] lives in [country0]. The [occupation1] lives in [country1]. The person living in [country0] is [gender0]. The [occupation1] living in [country1] is [gender1].

To ensure that the probing and prompting methods are detecting binding, and not relying on short cuts such as sentence order, for half of the contexts we swap the order of the last two sentences.

We compare probing and prompting at predicting correctly the gender of an occcupation mentioned in the context. For prompting, we prompt the model with "The gender of the [occupation] is", and take the gender with the higher log probability to be the model's answer. For probing, we take the gender token with higher binding similarity to the occupation token to be the probe's answer.

To evaluate these two methods, we showed both accuracy and calibrated accuracy. Accuracy is the fraction of the time that asking for the gender of an occupation in the context returns the correct answer. Calibrated accuracy requires more explanation. Language models sometimes exhibit innocuous but systematic preferences when evaluated in a forced-choice setting. For example, it might encounter the "male" token a lot more frequently than the "female" token, and so it might output "male" over "female" regardless of what the context or even the bias in the occupation in-dicates. A common practice is to calibrate the log probabilities (Robinson et al., 2022). We do so by subtracting the mean log probabilities in paired responses. Specifically, let $v_0, v_1 \in \mathbb{R}^2$ be the log probabilities over the male and female tokens when queried with occupation 0 and occupation 1. The calibrated log probabilities for occupation 0 is $v_0 - (v_0 + v_1)/2$, and that for occupation 1 is $v_1 - (v_0 + v_1)/2$. After obtaining the calibrated log probabilities, we apply the same decision rule as before, i.e. we choose the gender with the higher calibrated log probability as the answer. The same procedure can be applied to the binding similarities to calibrate probing. However, we find that calibration does not significantly change the accuracy of either method.

## I  QUANTITATIVE POSITION AND ORDER ANALYSES

In this section, we perform a more thorough analysis of the effectiveness of propositional probes as we vary the position and order of the name and attribute tokens. Broadly speaking, the model could utilize several features to inform its representation of how information is bound: the model could represent binding as semantically conveyed by the context, or rely on spurious features such as position information (i.e. nearby tokens are bound) or order information (i.e. names appearing in an order are bound to attributes in the same order.)

Overall, we find that the binding subspace we extract is not sensitive to position, but is affected partially by order. However, we note that the model's prompting performance becomes fragile when information is presented in unusual order, suggesting that the binding subspace we extract is influenced by order because the model's own representations are in fact influenced by order. Further, we find that despite the partial susceptibility to order, the propositional probes still outperform prompting in the adversarial settings.

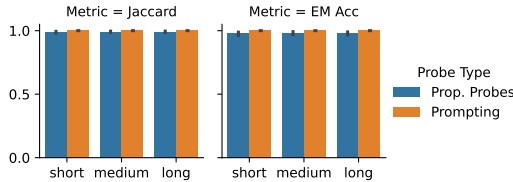

Figure 11: Propositional probes do not degrade with different lengths between name and attribute tokens.

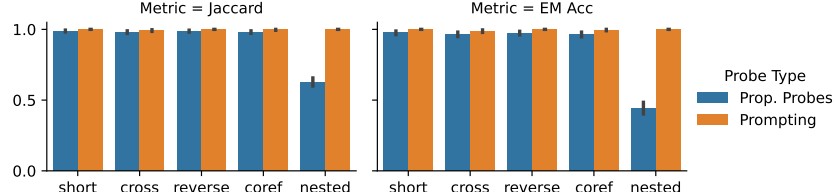

Figure 12: Propositional probes do well in all data orderings except for nested.

### I.1 POSITION ANALYSIS

As suggested by anonymous reviewers, we construct datasets of varying lengths between the entity and attribute positions. Specifically, we collect 20 noun phrases such as "dedicated advocate", 20 verb phrases such as "cultivates rare plants", and compositionally create contexts of the form: "Alice, a dedicated advocate who cultivates rare plants, lives in Germany. Bob lives in France". We call these contexts the LONG dataset. The MEDIUM dataset is similar, but does not contain verb phrases, e.g. "Alice, a dedicated advocate, lives in Germany. Bob lives in France.", and the SHORT dataset contains neither noun phrases nor verb phrases. As before, we sample random names and countries (as well as noun and verb phrases if applicable) to populate these templates, obtaining datasets of 512 input contexts.

Fig. 11 shows that propositional probes do not degrade in LONG or MEDIUM datasets as compared to SHORT. We additionally experimented with the version where both sentences in LONG have the long sentence structure, instead of just the first, and did not find a difference.

### I.2 ORDER ANALYSIS

In this section, we construct templates in which the order of the names and entity tokens is varied. To reduce clutter, we introduce a notation that represent names as capital letters A or B, and their corresponding countries as numbers 1 or 2. Then, the order of a context could be written succinctly as strings such as "A1B2". We also give each template a readable, informative name.

We list the templates and their examples below:

1. **series** (A1B2): "Alice lives in France. Bob lives in Germanay."
2. **cross** (AB12): "Alice and Bob live in France and Germany respectively."
3. **reverse** (A12B): "Alice lives in France. Germany is where Bob lives."
4. **coref** (AB12): "Alice and Bob are friends. The former lives in France. The latter lives in Germany."
5. **nested** (AB21): "Alice and Bob are friends. The latter lives in Germany. The former lives in France."

We find that propositional probes are robust to all data orderings we studied except for nested (Fig. 12). On the nested dataset, probes have an EM accuracy of 44%. If the binding is done independently and at random, we expect the accuracy to be 25%. If the binding is done based on order, we expect the accuracy to be 0%. We thus interpret the 44% as saying that the the binding subspace captures both order information and the true semantic binding information.

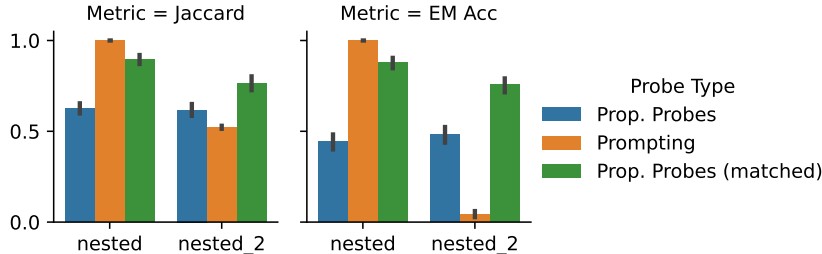

Figure 13: Constraining propositions to unique entities improves accuracy on both nested datasets ("matched" probes). A quirk in the language model makes the prompting strategy fail catastrophically on the second nested dataset.

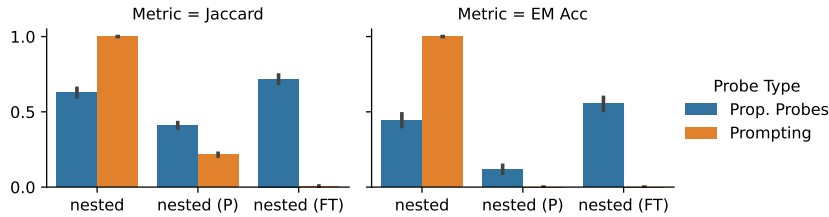

Figure 14: In the adversarial prompt injection (P) and backdoor (FT) versions of the nested dataset, we find that probes still outperform prompting.

We perform additional analyses to support this conclusion. First, we find that of the 56% where the probe outputs the wrong set of propositions, in 78% of these cases the probe assigns both countries to the same name. This suggests a continuous picture where the order information is in fact captured by our binding subspace, just that in some cases it is sufficiently strong enough to influence the decision boundary, and in others it is not. We find that a simple change where the propositions are constrained to bound to unique entities improves the accuracy to 76% (Fig. 13).

Further, there is evidence that the model itself relies on shortcuts such as order information to capture binding. One evidence comes from an alternate nested dataset we constructed, **nested_2** (AB21), that looks like: "Alice, unlike Bob who lives in Germany, lives in France." We find that on this dataset, our prompting strategy fails catastrophically, whereas the probes have similar performance to the original nested dataset **nested** (Fig. 13). Our error analysis indicates that prompting fails because the model is confused about the order of the entities in the context, and tends to say that both the first name and the second name in the context are "Alice". This suggests that in the nested order (AB21), the model's internal representations of binding may be fragile and conflated with order.

Finally, even on the original nested dataset where prompting does not suffer from the catastrophic failure, and using the original propositional probes algorithm that does not enforce unique entities, we find that on the adversarial settings the probes still outperform prompting (Fig. 14).

Similarly, in the gender bias set up, we split the dataset into contexts in which binding is in the series order (AB12) and the nested order (AB21), and find that while probes perform worse in nested order than in series order, they still outperform prompting when the occupations are anti-stereotypical (Fig. 15).

## J    LLAMA RESULTS

In this section, we show results for the Llama-2-13b-chat model, which is the instruction-tuned version of the base Llama-2-13b model. Our results are mostly similar.

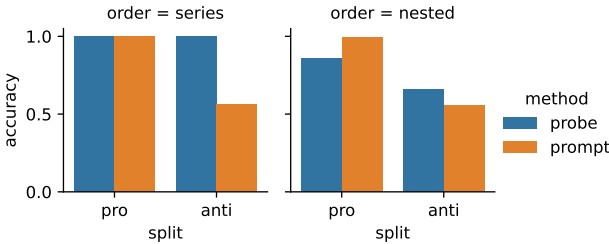

Figure 15: In the gender bias set up, the probe accuracy does degrade on the nested ordering (AB21) compared to the series ordering (AB12), but still outperforms prompting in both cases when examples are anti-stereotypical.

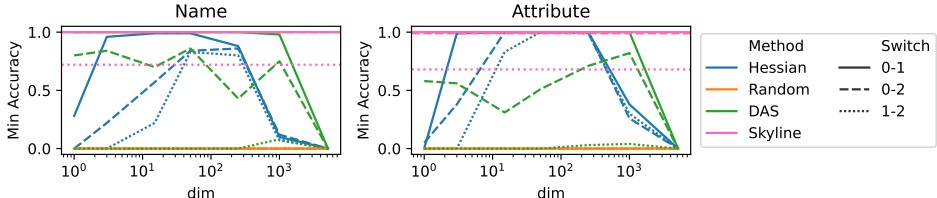

Figure 16: (Llama) The accuracy of swapping binding information in name (attribute) activations by projecting into $U_{(k)}$ ($V_{(k)}$) against $k$ in a context with 3 names and 3 attributes. We test the subspaces from the Hessian (blue), a random baseline (orange), and a skyline subspace obtained by estimating the subspace spanned by the first 3 binding vectors. We perform all 3 pairwise switches: 0-1 represents swapping the binding information of $E_0$ and $E_1$ ($A_0$ and $A_1$), and so on.

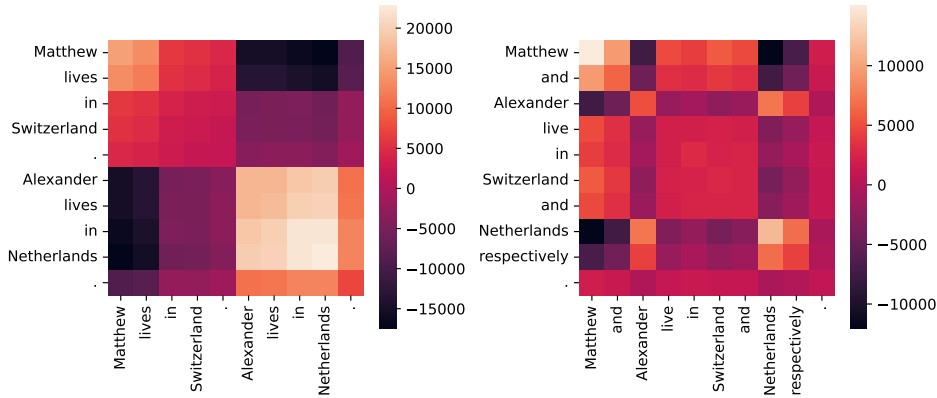

Figure 17: (Llama) Similarity between token activations under the binding similarity metric $d(\cdot, \cdot)$ for two-entity serial (**left**) and parallel (**right**) contexts.

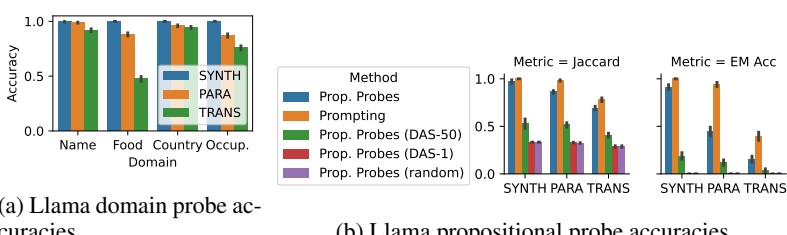

(a) Llama domain probe accuracies

(b) Llama propositional probe accuracies

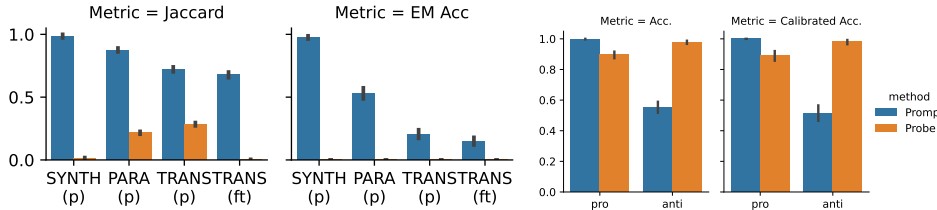

(a) Llama prop. probes in prompt injection (p) and dataset poisoning (ft) settings

(b) Llama prop. probes vs prompting for gender bias.

