# OpenReview forum: "Monitoring Latent World States in Language Models with Propositional Probes"
_ICLR.cc/2025/Conference — ICLR 2025 Spotlight_

### Official Review · Reviewer_vPtB · 2024-10-31

**Soundness:** 3
**Presentation:** 3
**Contribution:** 3
**Rating:** 6
**Confidence:** 3

**Summary:**

The paper proposes a method to extract latent world states in the form of propositional probes. They form predicate-argument-argument triples for multiple domains. They propose a method based on a Hessian-based algorithm in order to identify the binding subspace. They evaluate the propositional probes in both standard and adversarial settings. For the adversarial setting they find that the propositional probes stay more faithful to the input.

**Strengths:**

The paper is well-written and comes with multiple contributions. The contributions include the use of propositional probes, the definition of the hessian-based algorithm and the confirmation of two hypotheses - that propositions can be decided from internal activations and that these propositions are faithful to the input context.

**Weaknesses:**

- I would have liked to see a stronger focus on the adversarial experiments, in the paper. In particular, a deeper analysis on why probes remain faithful and how backdoor attacks and prompt injection could be prevented using your method.
- The synthetic dataset setup seems very simplistic and could have been made more true to real life use, such as by using paragraphs of existing texts and extracting propositions from them.

**Questions:**

- Why did you go from proposition to text and not the other way around: use existing text (from the wild) and generate propositions from it?

---

> ### Author Response · Authors · 2024-11-23
>
> Thank you for the thoughtful review. We are happy that you found our paper "well-written" and that it "comes with multiple contributions".
>
> > Why did you go from proposition to text and not the other way around: use existing text (from the wild) and generate propositions from it?
>
> Our main intuition is that going from structured representations (propositions) to unstructured representations (text) is easier than in the opposite direction. Further, there are some practical challenges involved in generating propositions from text-in-the-wild:
>
> - Not every piece of internet text fits the confines of the 4 domains that we have
> - Labels will not be balanced: probes can potentially cheat, e.g. by predicting the nationality of people from their names.
>
> This overall paradigm is related to the use of "minimal-pairs" in designing psychology experiments, which has been used in creating ML benchmarks such as EWOK (Ivanova, et al, 2024) and BigToM (Gandhi et al, 2024).
>
> However, we agree that if propositional probes are scaled to sufficiently diverse domains, text-in-the-wild can be a good downstream evaluation; it is still important to use a synthetic, balanced dataset as the main evaluation.
>
>
> Gandhi, Kanishk, et al. "Understanding social reasoning in language models with language models." Advances in Neural Information Processing Systems 36 (2024).
>
> Ivanova, Anna A., et al. "Elements of World Knowledge (EWOK): A cognition-inspired framework for evaluating basic world knowledge in language models." arXiv preprint arXiv:2405.09605 (2024).

---

> > ### Comment · Reviewer_vPtB · 2024-11-25
> >
> > Thank you for your response, this clarified my question about your synthetic dataset creation!

---

### Official Review · Reviewer_zN2i · 2024-11-02

**Soundness:** 4
**Presentation:** 3
**Contribution:** 4
**Rating:** 8
**Confidence:** 4

**Summary:**

The paper introduces a new probing technique, called propositional probes. Such probes are functions with two arguments both of which are language model representations. When applied to entities in the probe's domain, the output of the function is a symmetric metric which is expected to be high if the corresponding tokens are bound.
A Hessian-based algorithm is proposed to find a low-rank bilinear propositional probe. The algorithm starts with a way to query the language such that giving the correct answer depends on the ability to identify if entities are bound. In the paper's experiments, the language model is asked to repeat some relational information provided in-context (e.g. which country does entity0/entity1 live in). However, the representations of the two entities are set to their midpoint, such that the Hessian reveals how the representations would have to change in order to accurately represent their binding. After the Hessian is calculated, SVD is applied and only the top k-dimensional subspace is kept.
To evaluate this algorithm, 'interchange interventions' are performed where the positions in the identified subspace of two (out of three) entity representations are swapped. When the model is queried, it reports the 'wrong' entity with close to perfect accuracy for some values k. The binding strength is also visualized for some example inputs.
Further evaluations demonstrate that the probe match prompting performance in ordinary setting, and outperform it in adversarial settings.

**Strengths:**

- A new type of probe and corresponding algorithm to construct them are presented, this method is likely to be very useful to the interpretability community. Propositional probes will allow probing LLMs for the ability to represent entities as standing in certain relations to one another.
- The subspace identified is clearly shown to be causally implicated.
- Probes are shown to outperform prompting in adversarial setups.

**Weaknesses:**

- Side effects of the interventions are not investigated, it would be great to evaluate how much performance is/isn't lost, as an indication of how precise the interventions are.
- Results are limited to one model.

**Questions:**

- I noticed that the number of values of k that were evaluated is not super high, why is that? Is Hessian-based algorithm computationally intensive?
- Separate probes are learned for each domain, but every domain contains only one predicate, do you have any sense of how well propositional probes might generalize across predicates? Is there a good way to quantify how different the probes for each domain are?
- Do you think anything be gleaned from the singular values of H? Do they correlate with the accuracies in Figure 4 at all?

---

> ### Author Response · Authors · 2024-11-22
>
> Thank you for the thoughtful review. We are happy you felt that the "method is likely to be very useful to the interpretability community".
>
> > Results are limited to one model.
>
> We have run the experiments in the main paper on the Llama-2-13b-chat model, and put the results in a new appendix J. Broadly speaking, the results are very similar to the Tulu-2-13b model we used (which may not be surprising, because they are both finetuned from the same base model.)
>
> > I noticed that the number of values of k that were evaluated is not super high, why is that? Is Hessian-based algorithm computationally intensive?
>
> To compute the Hessian requires $d$ backward passes, where $d$ is the dimension of residual stream. For most open weight models this is around 1000-8000, which takes a few hours to run on A100 gpus. In principle the backward passes can be batched, but our attempts to use the torch functional APIs to achieve this have not been successful.
>
> > Separate probes are learned for each domain, but every domain contains only one predicate, do you have any sense of how well propositional probes might generalize across predicates? Is there a good way to quantify how different the probes for each domain are?
>
> At present, our results in Table 2 (right) show that of the four domains, (names, foods, countries, occupations), names probes generalize the best from the synthetic training data to paraphrased and translated data, whereas food probes generalize the worst. One reason why this might be the case is that representations for food items might be different in different contexts, whereas proper nouns like names are more straightforward. For example, 'apple' could connote the fruit, the tech company, or even New York City. In our paper, we quantify the performance of the probes with their generalization behavior from simple synthetic dataset to diverse paraphrased datasets, but there could be more systematic ways of comparing domain probes that directly study their representations.
>
> > Do you think anything be gleaned from the singular values of H? Do they correlate with the accuracies in Figure 4 at all?
>
> Yes, we think the singular values are important. The accuracies in Fig. 4 are obtained by sorting the singular values in descending order, and then taking the top $k$ singular vectors as the intervention subspace. If we were to sort in a random order, and take the top $k$ singular vectors (i.e. taking $k$ random vectors), we would have 0% accuracy.
>
> We also took a look at the magnitudes of the singular values. The singular values decay rapidly: the highest singular value is as high as ~250, and decays to ~10 by the 50-th vector, to ~1 by the 200-th, and to ~0.1 by the 1000-th. Since there are 4096 hidden dimensions, most of the singular values (> 3/4) are less than one-hundredth of the 50-th singular value, which is our cut off for the binding subspace.

---

> > ### Comment · Reviewer_zN2i · 2024-11-26
> >
> > Thank you for you reply. Great that you ran the experiments for an additional model. And thank you for answering my questions. I think it would be great if your answer w.r.t. implementation and performance details like the nr. of passes could be provided in the appendix as well.
> >
> > Another thing that might be fun to try if you find the time: project the top singular vectors into vocabulary/token space a la logit lens. Might give some insight into what the different singular vectors contribute.

---

### Official Review · Reviewer_JBEz · 2024-11-03

**Soundness:** 3
**Presentation:** 3
**Contribution:** 3
**Rating:** 8
**Confidence:** 4

**Summary:**

This paper proposes a method for finding a low-dimensional linear subspace of an LM's activation space which, so the main claim of the paper, encodes binding information.

While *binding* is a very broad and complex concept (see Greff et al., 2020, https://arxiv.org/abs/2012.05208 for a recent overview), in this paper binding refers to the process by which textual mentions of two or more entities and attributes that participate in a certain relation are bound together into a more abstract representation of that relation. For example, understanding the text "Alice lives in Laos" entails recognizing "Alice" and "Laos" as mentions of entities and then forming an abstraction of their relation that can be expressed as a proposition like LivesIn(Alice, Laos). On the representational level in a neural network this requires creating internal representations of the entities in question, and then performing some transformation on those representations that signals to subsequent layers that these two representations are "bound together".

The main hypothesis of the paper is that this transformation can be seen as a function that takes two entity representations x and y as input and outputs their "binding strength" F(x, y), i.e., a score that quantifies whether the two entities are bound or not. Assuming that F is bilinear in the entity representations x and y, the authors propose a method to estimate F via its Hessian matrix H. If the binding subspace is low-dimensional, then F and the Hessian H should be low rank, which motivates the authors to analyze the rank-k truncated SVD of H. By measuring the efficacy of interchange interventions as a function of k, the authors find that a k=50 dimensional subspace mediates binding, i.e., when manipulating activations in this subspace model output changes accordingly. For example, given the input "Alices lives in Laos. Bob lives in Peru." one can make the LM say "Bob lives in Laos" by intervening on activations in this low-dimensional subspace.

Having developed the machinery to probe an LM for internal representations of propositions, the paper demonstrates several use cases for analyzing discrepancies between the model's internal representations and its output, finding cases in which the model appears to internally represent a proposition but generates output that is inconsistent with it.

**Strengths:**

**Originality:** The paper proposes a novel method for identifying a low-dimensional subspace which appears to causally mediate binding behavior in language models. Compared to the rather indirect evidence seen in prior work (Feng & Steinhardt, 2023), the present submission directly identifies this subspace, which results in a much greater degree of manipulability and interpretability.

**Quality:** The method is well-motivated (§5.2) and -- at least on the data used -- works well empirically (§6.1). The qualitative analysis (Figures 5, 7, 8 and related discussion) nicely illustrates similarity in the low-dimensional dimensional subspace, as well as the limitations of the method.

**Clarity:** The paper is structured well, is clearly written and flows naturally.

**Significance:** Binding is generally believed to be an essential component in the formation of internal/situation/world models. As such, any progress towards understanding if/how language models perform binding on a representational level is an important contribution.

**Weaknesses:**

**Edit after author response:** The authors thoroughly addressed all issues by conducting experiments that disentangle position, order, and "true" binding. I've raised my scores accordingly: soundness 2 -> 3, overall rating 6 -> 8

---
**Weaknesses before revision below. These issues are now sufficiently addressed in the appendix of the revised manuscript:**

The paper does not do enough to rule out an alternative, simpler hypothesis that could potentially explain the results. Concretely, it appears possible that, due to the highly regular nature of the data, the low-dimensional subspace claimed to encode binding information primarily encodes positional information or order information. The running example "Alices lives in Laos. Bob lives in Peru." has highly regular structure with fixed positional offsets between person and country names, so it is conceivable that the proposed method actually identifies a "position subspace" or "entity/attribute order subspace" and that interchange interventions claimed to modify binding information in fact modify positional or order information. The paper takes two steps into the direction of ruling out this alternative explanation, but I do not believe that they are sufficient:
1. Using a LM to rewrite the template-generated texts into short story-like paraphrases. My concern here is that it is unclear how much of the original regularity remains in the paraphrases and how variations in the paraphrases relate to probe performance in Table 2. Since the probe performance exact match metric on the paraphrase is much lower than on the template-based data, it is possible that the probe works best on the paraphrases that are structurally closer to the templates and drops as the paraphrases become more varied and creative. An additional analysis looking at, say, probe performance as a function of token distance between entities and attributes in the paraphrases could provide evidence for or against position being encoded in the identified low-dimensional subspace.
2. A qualitative comparison in which position and order are varied ("parallel" setting in Figure 5, coreference examples in Figure 7). While encouraging, these are only a few qualitative examples of representational similarities. Here, systematic, quantitative experiments would go a long way towards ruling out alternative explanations. Data for such experiments could be relatively easily generated by varying position and order in the templates, e.g., "Alice lives currently and has always lived Laos. Bob lives in Peru", which varies the token distance between the bound arguments or "Alices lives in Laos. Peru is where Bob lives.", which swaps the order of arguments. If the authors can show that the subspace mediates binding in a similar manner, this would make a much stronger justification for calling it a "binding subspace"

**Questions:**

My main concern and suggestions on how to alleviate it is given in the weaknesses. If the authors can present evidence that helps rule out the positional/order explanation I'm more than happy to raise my score.

---

> ### Author Response · Authors · 2024-11-22
>
> Thank you for the thoughtful review. We share your belief that identifying binding in representation space is an important contribution.
>
> The main weakness you highlighted was that: "The paper does not do enough to rule out an alternative, simpler hypothesis that could potentially explain the results." Specifically, you are concerned that instead of the true binding vector, the binding subspace could be capturing spurious information such as position or order.
>
> Based on your suggestions, we conducted a thorough quantitative investigation (Appendix I). Our main finding is that the extracted binding subspace is not influenced by position, but captures both the true binding vector and the order so that performance is degraded in a particular ordering of names and countries. However, despite the binding subspace's partial susceptibility to order, the propositional probes still outperform prompting in the adversarial settings.
>
> Here, we briefly summarize the key experiments, and leave the details to Appendix I.
>
> - We create templates of increasing distance between the first name and country, and found that propositional probes perform consistently well. We therefore conclude our binding subspace is not influenced by position.
> - We create 5 templates corresponding to different orderings of names and countries. These 5 orderings include the one you proposed "Alice lives in Laos. Peru is where Bob lives", which we call the __reversed__ ordering.
> - 4 of the 5 orderings, including the proposed __reversed__ ordering, show consistent performance.
> - 1 particular ordering shows a degradation in the probe's performance. We call this ordering the __nested__ ordering, and it looks like this: “Alice and Bob are friends. The latter lives in Germany. The former lives in France.”
> - This degradation is significant but not total. 44% of the time, the probe still returns the correct propositions. 50% of the time, the probe assigns both countries to the same name. If the binding subspace had been capturing only order information, but not binding, we would expect the propositions to be wrong 100% of the time. In contrast, the results are more consistent with the interpretation that the binding subspace captures both binding and also spurious features corresponding to order.
> - However, if we apply the nested ordering to the adversarial settings (prompt injection, dataset poisoning, and gender bias), our probes still out perform prompting
> - There are signs that the language model itself may conflate order and binding. In an alternate binding template that looks like "Alice, unlike Bob who lives in Germany, lives in France.", we find that prompting fails catastrophically because the model thinks that both the first name and the second name in the context are "Alice".
> - The degradation to propositional probes can be ameliorated by a small change to the composition algorithm: constraining the proposed propositions to have distinct entities
>
> Overall, we are grateful for your suggestions. Our experiments based on your suggestions imply that binding may be entangled with order, and our methods can be further improved by attempting to disentangle them. Nonetheless, our methods at present are still robust enough to outperform prompting in the adversarial settings.

---

> > ### Comment · Reviewer_JBEz · 2024-11-26
> > **convincing experiments, raised scores**
> >
> > Thank you for conducting these experiments in such short time! I think they're very convincing and your interpretation of the results makes sense. I've raised my scores accordingly.

---

### Official Review · Reviewer_RZaJ · 2024-11-03

**Soundness:** 3
**Presentation:** 3
**Contribution:** 3
**Rating:** 8
**Confidence:** 3

**Summary:**

The paper studies how LLMs might encode propositions stated in the context, like "Greg is a nurse. Laura is a physicist.", by looking at the activations associated with the Greg/nurse tokens, and trying to identify "propositional probes" through a "binding subspace" of these vectors which are aligned when the proposition holds.

They use a synthetic set with 4 domains (people, countries, foods, occupations), each with a set of non-overlapping entities (from 14 to 60 per domain). They define a somewhat heuristic "domain identifier" probe to pick up tokens associated with each entity, and then (main novelty) use a low-rank Hessian approach to identify these binding subspaces.

There is analysis for how effective these subspaces are in changing the binding between entities (e.g., to make Greg the physicist after the context above, when answering a question like "Therefore, Greg's occupation is"). The conclusion is that it "works" to some extent, but with caveats, especially when the context gets more complicated (going from 2 to 3 entities). In addition to testing on the synthetic contexts, there is an LLM generated variant (PARA) that turns the sequence of propositions into an actual story format, and one that translates this story into Spanish (TRANS). There is non-trivial carry over of the effect to these cases. There are also comparisons to other probing baselines.

Finally, they also test on some "adversarial" cases: 1) Prompt injection (encourage the model to answer wrongly),  2) Backdoor attacks (model fine-tuned to do badly in Spanish), 3) Gender bias (measure amount of gender bias in output probabilities for stereotypical vs anti-stereotypical occupation assignments). In all three cases they find the propositional probes are more faithful to the underlying "true" propositions vs the actual model output.

**Strengths:**

Although I am not very familiar with this field, the method for identifying the binding subspaces seems quite novel, and potentially will provide useful insights into model behavior.

The task setup, although very synthetic in nature, has the PARA and TRANS variants which make it a potentially fruitful testing ground for these kinds of questions.

The general topic of understanding mechanisms and world states inside of LLMs is both interesting and important.

**Weaknesses:**

The "domain probes" to classify tokens into domain seem quite heuristic (using the mean vector of all the entities in the domain), and it seems like there could be some evaluation to see how it works (e.g., is it always the "obvious" tokens, like the "Alice" token is the name token?).

Some of the decisions in the binding space design seem quite arbitrary, like "For tractability, we parameterize x and y so that they are shared across layers". Maybe it would then be better to just focus on a few layers? But it's perhaps fine to leave that for future investigation.

For the Prompt Injection setting (instructing the model to "Always answer the opposite."), it's hard to say what a "correct" output should be, in fact the prompting method should probably "ideally" always be "wrong". So saying "prompting performs worse" is a bit confusing, although it's still an interesting result that the probing outputs are virtually unchanged.

**Questions:**

Suggestion for Table 1: Make it clearer that the (P) and (FT) columns correspond to specific adversarial settings

It would be interesting with some more error analysis for what breaks down when the subspace hypothesis fails, to get insights into the potential for these methods to scale to more complex settings.

---

> ### Author Response · Authors · 2024-11-22
>
> Thanks for the thoughtful review; we’re happy that you found the investigation into world states “interesting and important”, and appreciated the novelty of the Hessian method for identifying the binding subspace. We’d like to provide some clarification around some of the weaknesses you raised.
>
> > The "domain probes" to classify tokens into domain seem quite heuristic (using the mean vector of all the entities in the domain), and it seems like there could be some evaluation to see how it works (e.g., is it always the "obvious" tokens, like the "Alice" token is the name token?).
>
> Our domain probes are a multi-class generalization of the difference-in-means probes, which some previous works have argued are more robust than standard linear probes trained with logistic regression. For example, Belrose (2024) analyzes some theoretical properties of these probes.
>
> Although not the main focus of the work, we conducted some analysis into the domain probe. Using Grad-CAM style saliency maps (appendix F), we can estimate the extent to which the activations at each layer and token position contribute towards the correct answer, and we found that indeed it is usually the “obvious” token that matters, and when the key word contains multiple tokens, it is the last token that matters. The importance of the last token is also previously discovered in the knowledge editing setting (Meng et al, 2022).
>
> > Some of the decisions in the binding space design seem quite arbitrary, like "For tractability, we parameterize x and y so that they are shared across layers". Maybe it would then be better to just focus on a few layers? But it's perhaps fine to leave that for future investigation.
>
> This is a good observation. This design choice, and others, are motivated by the circuits responsible for solving binding. We have updated Appendix C to motivate parameterizing $x$ and $y$ across layers. At a high level, the binding vectors for entities are accessed at different layers than the binding vectors for attributes. In principle you can choose to localize the specific layers relevant for entities and attributes, and then perform the injection for those layers — we expect this idea to further improve the accuracy of the binding subspace, at the cost of having more complexity and hyperparameters to tune.
>
> > For the Prompt Injection setting (instructing the model to "Always answer the opposite."), it's hard to say what a "correct" output should be, in fact the prompting method should probably "ideally" always be "wrong". So saying "prompting performs worse" is a bit confusing, although it's still an interesting result that the probing outputs are virtually unchanged.
>
> We can see why this is confusing. There are two different ways of framing this that can justify answering the wrong answer as the undesired output. First, from the security point of view, prompt injections are malicious parts of the input that poisons the input context so that the model stops performing as we expect. For example, consider an LLM agent browsing the web on a user's behalf. If it encounters a piece of text that asks it to start misleading the user, and follows the instruction to mislead the user, then we would ideally want some way to monitor the LLM agent when this happens.
>
> The second way of framing it is that we are interested in latent world models in language models. Perhaps, despite being instructed to lie, the model will still first form a coherent world state internally, and then modify its output based on this world state. Our results show exactly this.
>
>
> Belrose, Nora. "Diff-in-means concept editing is worst-case optimal: Explaining a result by Sam Marks and Max Tegmark, 2023." URL https://blog. eleuther. ai/diff-in-means (2024).
>
> Meng, Kevin, et al. "Locating and editing factual associations in GPT." Advances in Neural Information Processing Systems 35 (2022): 17359-17372.

---

> > ### Comment · Reviewer_RZaJ · 2024-11-27
> > **Response to authors**
> >
> > Thank your for your detailed responses, and additions to the paper which do increase its value in my eyes. I have raised my overall score from 6 to 8.

---

### Author Response · Authors · 2024-11-22

We thank all four reviewers for their helpful feedback. We are happy that reviewers found the binding problem important especially in the context of understanding internal world models in language models (RZaJ, JBEz), and that the method is "novel" (RZaJ), "original" (JBEz), and "likely to be very useful to the interpretability community" (zN2i).

The reviewers also pointed out several areas which we could work on. We address them in individual comments.

---

### Meta-Review · Area_Chair_YE3P · 2024-12-18

**Metareview:**

This paper proposes a method to interpret language models by extracting logical propositions that they claim represent the internal state of the model, by discovering bindings in a low-dimensional linear subspace of the model. The subspace discovery algorithm and the propositional probes were evaluated independently; the latter was done in standard and adversarial settings. While the model can generate falsehoods, these propositions may still hold; this was shown to hold for different adversarial settings.

**Strengths**: The idea is novel and the algorithm can be, in principle, generalized to extract other phenomena, including more complex propositional logic. Experimental results in the standard settings were sound and convincing.


**Weaknesses**: Like some of the reviewers, I was a little underwhelmed and baffled by the experimental findings in the adversarial setting:  the model does engage in "deceitful" behavior despite having a "correct" internal world representation which makes it somewhat antithetical to the original hypothesis of the paper. For instance, what are the implications when models exhibit behavior unfaithful to its internal state? Does this mean that the internal world representation is weak?

**Reason for acceptance**: See strengths; contributions of this work other than the adversarial settings are quite interesting and potentially useful, and the work has its merits.

**Additional Comments On Reviewer Discussion:**

Reviewers' objections to the work included the synthetic nature of the task setup - this was addressed by the authors by motivating their work from the use of minimal pairs common in theory of mind experiments. Some issues regarding design choices were also pointed out which the authors resolved by presenting additional experiments. There were also some issues with the adversarial setting; the authors' response seems to have addressed most reviewers' objections.  Regardless, I agree with the reviewers with other strengths of this work.

---

### Decision · Program_Chairs · 2025-01-22

Accept (Spotlight)